# Initial Use of 100% but Not 60% or 30% Oxygen Achieved a Target Heart Rate of 100 bpm and Preductal Saturations of 80% Faster in a Bradycardic Preterm Model

**DOI:** 10.3390/children9111750

**Published:** 2022-11-15

**Authors:** Mausma Bawa, Sylvia Gugino, Justin Helman, Lori Nielsen, Nicole Bradley, Srinivasan Mani, Arun Prasath, Clariss Blanco, Andreina Mari, Jayasree Nair, Munmun Rawat, Satyan Lakshminrusimha, Praveen Chandrasekharan

**Affiliations:** 1Department of Pediatrics, Jacobs School of Medicine & Biomedical Sciences, University at Buffalo, Buffalo, NY 14260, USA; 2Department of Pediatrics, Boston Children Hospital, Harvard Medical School, Boston, MA 02115, USA; 3Department of Pediatrics, Division of Neonatology, University of California Davis School of Medicine, Sacramento, CA 95817, USA

**Keywords:** bradycardia, supplemental oxygen, saturations, heart rate

## Abstract

**Background:** Currently, 21–30% supplemental oxygen is recommended during resuscitation of preterm neonates. Recent studies have shown that 58% of infants < 32 week gestation age are born with a heart rate (HR) < 100 bpm. Prolonged bradycardia with the inability to achieve a preductal saturation (SpO_2_) of 80% by 5 min is associated with mortality and morbidity in preterm infants. The optimal oxygen concentration that enables the achievement of a HR ≥ 100 bpm and SpO_2_ of ≥80% by 5 min in preterm lambs is not known. **Methods:** Preterm ovine model (125–127 d, gestation equivalent to human neonates < 28 weeks) was instrumented, and asphyxia was induced by umbilical cord occlusion until bradycardia. Ventilation was initiated with 30% (OX30), 60% (OX60), and 100% (OX100) for the first 2 min and titrated proportionately to the difference from the recommended preductal SpO_2_. Our primary outcome was the incidence of the composite of HR ≥ 100 bpm and SpO_2_ ≥ 80%, by 5 min. Secondary outcomes were to evaluate the time taken to achieve the primary outcome, gas exchange, pulmonary/systemic hemodynamics, and the oxidative injury. **Results:** Eighteen lambs (OX30-6, OX60-5. OX100-7) had an average HR < 91 bpm with a pH of <6.92 before resuscitation. Sixty seven percent achieved the primary outcome in OX100, 40% in OX60, and none in OX30. The time taken to achieve the primary outcome was significantly shorter with OX100 (6 ± 2 min) than with OX30 (10 ± 3 min) (* *p* = 0.04). The preductal SpO_2_ was highest with OX100, while the peak pulmonary blood flow was lowest with OX30, with no difference in O_2_ delivery to the brain or oxidative injury by 10 min. **Conclusions:** The use of 30%, 60%, and 100% supplemental O_2_ in a bradycardic preterm ovine model did not demonstrate a significant difference in the composite primary outcome. The current recommendation to use 30% oxygen did not achieve a preductal SpO_2_ of 80% by 5 min in any preterm lambs. Clinical studies to optimize supplemental O_2_ in depressed preterm neonates not requiring chest compressions are warranted.

## 1. Introduction

About half (58%) of the preterm neonates < 32 weeks gestation at birth are born bradycardic with a heart rate (HR) < 100 bpm [1]. In preterm neonates requiring respiratory support, increasing HR and preductal saturations (SpO_2_) are key indicators of successful resuscitation [2,3]. The evaluation of data from preterm neonates enrolled in randomized controlled trials (RCTs) has shown that failure to achieve SpO_2_ of 80% and a HR of ≥100 bpm increased the risk of morbidity and mortality [4]. The risk of mortality is 18 times higher in bradycardic, hypoxemic preterm neonates < 32 weeks gestation than in preterm neonates without bradycardia or hypoxemia [1]. Unless a preterm neonate requires chest compressions, the current recommendation by the International Liaison Committee on Resuscitation (ILCOR) is to initiate resuscitation with supplemental oxygen (O_2_) between 21 and 30% with subsequent titration based on target preductal SpO_2_ [2,5,6]. In preterm neonates with surfactant deficiency (respiratory distress syndrome (RDS)) and acidosis secondary to perinatal depression, the optimal initial supplemental oxygen use remains unclear. The use of higher oxygen concentration for the resuscitation of preterm neonates with RDS could lead to oxygen toxicity [7]. Our goal was to evaluate the use of different concentrations of initial supplemental O_2_ (30%, 60%, and 100%) on the incidence of HR of ≥100 bpm and SpO_2_ ≥ 80% by 5 min in preterm lambs. For this purpose, we utilized an ovine preterm model born with bradycardia (HR of <100 bpm) induced by umbilical cord compression that is equivalent to <28 week gestation preterm human neonate.

## 2. Materials and Methods

The study was approved by the Institutional Animal Care and Use Committee (IACUC approval-PED 10085N), University at Buffalo, Buffalo, NY, USA and followed the ARRIVE guidelines. Time-dated ewes (125–127 d, gestation, term ~147–150 d) were used in this study. The ewes did not receive antenatal steroids. After an overnight fast, the ewes were anesthetized, a caesarean section was performed, and fetal lambs were partially exteriorized. These fetal lambs were instrumented while in placental circulation as described previously [6]. Instrumentation included the placement of the right jugular and carotid lines for access, pressure monitoring, and blood draws. Ultrasonic flow probes (Transonics, Ithaca, NY, USA) were placed around the left carotid artery, left pulmonary artery, and ductal arteriosus to monitor blood flows. The lambs were intubated, and the lung fluid was drained by gravity.

Asphyxia was induced by umbilical cord occlusion until the heart rate (HR) reached <90 bpm. We targeted a HR < 90 bpm to ensure that the preterm lambs had a consistent HR < 100 bpm at onset of resuscitation. Once the target HR was achieved, the fetus was randomized to one of the three groups to receive supplemental oxygen: OX30-initiate resuscitation with 30% O_2_ and titrate after 2 min based on preductal SpO_2_, OX60-initiate resuscitation with 60% O_2_ and titrate after 2 min based on preductal SpO_2_, or OX100-initiate resuscitation with 100% O_2_ and titrate after 2 min based on preductal SpO_2_.

Oxygen titration based on preductal SpO_2_: We waited until 2 min to titrate O_2_ to mimic a clinical scenario because transferring a newborn to a radiant warmer for resuscitation, the placement of a pulse oximeter probe, and the acquisition of a signal takes time. The titration of O_2_ was proportional to the difference between the observed SpO_2_ and the target SpO_2_ and was performed every min. If preductal SpO_2_ is outside the target range, the inspired O_2_ will be adjusted every min as follows:(a)It will be adjusted by 10% if SpO_2_ is less than 10% outside the range (e.g., target SpO_2_ 70%, previous minute SpO_2_ 60%, then increase O_2_ by 10%) or(b)It will be adjusted by 20% if SpO_2_ is greater than 10% outside the range (e.g., target SpO_2_ 80%, previous minute SpO_2_ 50%, then increase O_2_ concentration by 20%).

Similarly, the supplemental O_2_ was decreased based on target SpO_2_ every min. In order to obtain an accurate and continuous recording of preductal SpO_2_, two preductal probes were placed—one on the tongue and another on the right upper limb. The positive pressure ventilation in all groups targeted a tidal volume of 7–8 mL/kg by adjusting the peak inspiratory pressure accordingly. The positive end expiratory pressure was set at 6–7 cm H_2_O. Using a respiratory monitor, the above parameters were continuously monitored along with end-tidal carbon dioxide (ETCO_2_) levels.

Our primary outcome was the frequency of composite of HR ≥ 100 bpm and SpO_2_ ≥ 80% by 5 min of resuscitation.

Our secondary outcomes were to evaluate (i) the time taken to achieve the primary outcome, (ii) the gas exchange, (iii) the pulmonary/systemic hemodynamics, and (iv) the oxidative injury.

We obtained data on preductal SpO_2_, left pulmonary blood flow, left carotid blood flow, and arterial blood gas. We calculated the oxygen delivery to the brain as shown previously [8,9]. For oxidative injury, we measured the oxidized to reduced glutathione ratio (GSSG/GSH) in blood samples in all three groups at 10 min and used this sample to calculate oxygen delivery to the brain.

Sample size calculation: From our previous studies, involving the asphyxiated preterm RDS lamb model, the difference in chance of achieving the primary outcome (HR ≥ 100/m and SpO_2_ 80%) between the proposed groups was 20 ± 10%. Using one way ANOVA and the statistical program SAS 9.4 (NC, USA) and considering the nature of the study, the sample size was calculated, assuming that the primary outcome is a single variable (HR ≥ 100/m and SpO_2_ 80%), and the group means of achieving primary outcomes are 20%, 40%, and 60%. We used contrast coefficients of −1, 1, and 1 to factor in differences between current recommendations of OX30 compared to experimental groups OX60 and OX100. A total of 15 lambs (N-5 in each group) were needed, which provided a power of 0.914 to obtain a difference in primary outcomes; with an N of 6 in each group, this power increased to 0.980. The Type I error probability for this calculation is 0.05.

Parametric data are presented as mean and standard deviation and analyzed by ANOVA. Non-parametric data are presented as median, interquartile range, and range and analyzed by the Kruskal–Wallis test. The significance was set at a probability of less than five percent, as mentioned previously.

## 3. Results

A total of 18 preterm lambs were randomized to OX30 (N-6), OX60 (N-5), and OX 100 (N-7) groups. The characteristics of these lambs are shown in Table 1, and there were no significant differences between the groups. The average HR at resuscitation was 90 bpm with a pH of 6.9, as shown in Table 1.

### 3.1. Primary Outcome

Initiating resuscitation with 30% oxygen (OX30) did not achieve the primary outcome in any of the asphyxiated preterm lambs. In OX60, 40% achieved the primary outcome, and in OX100, 67% achieved the primary outcome (Table 2), but this difference was not statistically significant. 

### 3.2. Time Taken to Achieve an HR ≥ 100 bpm and SpO_2_ ≥ 80%

The time taken to achieve an HR ≥ 100 bpm and SpO_2_ ≥ 80% was significantly lower with OX100 (6 ± 2 min) compared to OX30 (10 ± 3 min) (* *p* = 0.04-post hoc, ANOVA). With OX60, it took 8 ± 3 min to achieve the target (Table 2). 

### 3.3. Preductal Saturations, Supplemental Oxygen Use, Oxygenation, and Ventilation

#### 3.3.1. Preductal Saturations

The preductal saturations during the first ten minutes are shown in Figure 1. The preductal saturation was significantly higher with OX100 than with OX30 in the first 10 min. (* *p* < 0.01).

#### 3.3.2. Supplemental Oxygen Use

The supplemental use of oxygen during the first ten minutes of resuscitation is shown in Figure 2. The average supplemental oxygen was similar between the three groups (OX100—89 ± 22%, OX60—97 ± 7%, and OX30—92 ± 16%). If the target saturation was not achieved, as mentioned in the methodology, the supplemental oxygen was titrated (in-creased/weaned). Additionally, the median and ranges for the supplemental O_2_ in these groups were OX100—89 (21–100), OX60—100 (60–100), and OX30—100 (30–100), respectively.

#### 3.3.3. Arterial Oxygenation (PaO_2_)

The PaO_2_ values during the first 10 min after birth are shown in Figure 3. The arterial PaO_2_ values were significantly lower in the OX30 group. Despite initiation with 100% oxygen, the mean PaO_2_ values did not exceed 100 mmHg in the first 10 min in the OX100 group. 

#### 3.3.4. Ventilation

The ETCO_2_ levels in the first 10 min were OX30 44 ± 15, OX60 45 ± 24, and OX30 45 ± 16 mmHg, and these were similar between the groups. However, the corresponding PaCO_2_ in the groups OX30 110 ± 37, OX60 104 ± 28, and OX30 102 ± 16 mmHg were not different between the groups. In a surfactant-deficient asphyxiated model, without antenatal betamethasone, there is carbon dioxide retention. With ventilation perfusion mismatch in the absence of surfactant, there was a considerable difference between ETCO_2_ and PaCO_2_ in our model.

### 3.4. Hemodynamics

#### 3.4.1. Peak Pulmonary Blood Flow

The peak (systolic) pulmonary blood flow over the first 10 min of resuscitation is shown in Figure 4. Peak pulmonary flow was significantly lower during this period in the OX30 group.

#### 3.4.2. Peak Carotid Blood Flow

The peak CBF was not different between the OX30, OX60, and OX100 groups, as shown in Figure 5. 

### 3.5. Oxygen Delivery to the Brain

The calculated oxygen delivery to the brain in all three groups showed no difference in the first 10 min between the groups, as shown in Figure 6. 

### 3.6. Oxidized to Reduced Glutathione Ratio between the Groups

The GSSG/GSH ratio as shown in Figure 7 was not different at 10 min between the groups. 

## 4. Discussion

Extremely preterm neonates are surfactant-deficient, and optimizing oxygenation and ventilation is a critical step during resuscitation [2]. Increasing HR is the best measure of effective ventilation in depressed neonates. The optimal initial oxygen concentration during the resuscitation of preterm infants continues to be a subject of controversy [7,10]. It remains unknown if lower initial supplemental O_2_ concentration would be beneficial in an asphyxiated surfactant-deficient preterm neonate. However, there is a fine balance between achieving increased oxygenation and causing injury from oxidative stress because preterm infants have an immature defense system against oxidative injury [11]. The current recommendations from ILCOR/NRP are to initiate resuscitation with 21 to 30% oxygen, monitor preductal SpO_2,_ and titrate the supplemental O_2_ based on pre-specified SpO_2_ targets [5,6]. The importance of monitoring the HR and SpO_2_ in preterm neonates and its significance have been shown in two large meta-analysis. 

Oei et al. included 768, <32 weeks preterm infants from 8 RCTs in a meta-analysis and showed that after accounting for confounding factors, gestational age (GA), birth weight, and bradycardia (defined as HR < 100 bpm) at 5 min, the time taken to reach SpO_2_ of 80% was associated with higher mortality [4]. The recent individual patient meta-analysis by Kapadia et al. included 720 infants with GA of <32 weeks from 8 RCTs and showed that 58% infants were born with a HR of <100 bpm [1]. The study concluded that infants with prolonged bradycardia for >2 min and SpO_2_ < 80% at 5 min had 18 times higher odds of mortality. Both of these studies included preterm infants from low (≤30%) and high (≥60%) O_2_ RCTs. The incidence of bradycardia was similar in both of these groups. If preterm neonates are born depressed, could higher oxygen exposure during ventilation benefit or harm them? Small, randomized trials suggest that initiating resuscitation with 100% oxygen with a rapid wean increases diaphragmatic activity [12,13,14]. The hemodynamic mechanisms by which a higher supplemental O_2_ such as 100% would help transition these depressed neonates to achieve a HR of ≥100 bpm and SpO_2_ of ≥80% by 5 min is not well established. 

In our study, none of the preterm lambs with the use of the current recommendation of 30% achieved the primary outcome of HR of ≥100 bpm and SpO_2_ of ≥80% by 5 min. Even with initiating resuscitation with a supplemental 100% O_2_, only 67% of the depressed preterm lambs achieved the primary outcome by 5 min. These preterm lambs without prior exposure to antenatal steroids or the administration of surfactant have high alveolar-to-arterial oxygen gradients (A-a DO_2_). With 100% supplemental O_2_, the average arterial O_2_ tension (PaO_2_) was <100 mm Hg in the first 10 min. Perinatal depression, acidosis, and RDS can hamper the decrease in pulmonary vascular resistance and an improvement in pulmonary blood flow [8,15,16]. In our study, the use of 30% oxygen was associated with lower peak pulmonary blood flow than the other groups. This is also shown by the extent of the difference between ETCO_2_ and PaCO_2_ in the first 10 min. Transient alveolar hyperoxia induced by initiating resuscitation with 100% oxygen may promote pulmonary vasodilation compared to 30% oxygen [17]. 

Asphyxia, in any case, can lead to a cerebral inflammatory response and puts preterm infants at a higher risk of intraventricular hemorrhage (IVH), especially if higher supplemental O_2_ is used during resuscitation [18,19,20]. However, in our study, the O_2_ delivery to the brain measured using the left carotid blood flow was not different between the groups. The variability seen in the OX30 group could be secondary to the wide range of supplemental oxygen used in this group in an asphyxiated surfactant-deficient model. Again, it is tough to delineate whether IVH in preterm neonates is secondary to asphyxia with pronged bradycardia or secondary to high oxygen exposure in a clinical scenario. From previous clinical studies, we know that the use of 100% supplemental O_2_ leads to higher oxidative stress than the use of lower or 21% O_2_ [7,21,22,23,24,25]. In this ovine model, there were no differences noted with the three concentrations of O_2_ used during resuscitation on the oxidized to reduced glutathione ratio. In our preterm model, our oxidative stress marker (GSSG/GSH) ratio results are from oxygen exposure for the first 10 min only, which is titrated based on preductal saturations. This response could be different in term neonates as the antioxidant mechanisms mature in parallel to the lung growth and surfactant production that occur closer to term gestation. Term babies are more capable of handling higher oxygen exposure than preterm neonates. Rook et al. have shown in their study that the infusion of amino acids could lead to alterations in GSH levels [26]. Furthermore, subsequent oxygen exposure post resuscitation, especially in hypoxic preterm neonates, could lead to higher oxidative stress. Preterm neonates born with a HR of <100 bpm may need a higher initial oxygen concentration to be resuscitated based on our findings. Clinical studies that can measure the markers of oxidative injury immediately after the delivery room resuscitation may help us understand the potential ill-effects or benefits of high oxygen exposure in the setting of hypoxia/bradycardia.

There are several limitations to this study. The studies were conducted in a controlled environment by experienced personnel. Physiological differences in species may limit the translation of these findings to clinical trials. The preterm lambs were intubated before resuscitation, as compared to initiating ventilation with a mask in preterm human neonates. However, the physiology of an asphyxiated preterm neonate is mimicked by a lamb with fetal lung liquid, an open ductus arteriosus, and high pulmonary vascular resistance undergoing umbilical cord compression. The differences in oxidative stress markers between preterm and term model need further evaluation. To our knowledge, this is the first study to use a transitional asphyxiated preterm ovine model to understand the effect of O_2_ exposure and titration based on preductal SpO_2_ for inducing moderate bradycardia not requiring chest compressions. Our study also uses a robust protocol for O_2_ titration based on preductal SpO_2_, which could be used in future trials.

## 5. Conclusions

The use of 30%, 60%, and 100% supplemental O_2_ in a bradycardic preterm ovine model did not demonstrate a significant difference in the composite primary outcome of HR ≥ 100 bpm and SpO_2_ ≥ 80% by 5 min of resuscitation. The use of 100% supplemental O_2_ achieved the primary outcome faster, with an improvement in arterial oxygen tension and increased peak pulmonary blood flow with no difference in oxygen delivery to the brain or oxidative stress injury, compared to 60% and 30% O_2_. With more than 50% of preterm neonates born depressed with an HR < 100 bpm, additional translational and clinical trials are warranted to optimize initial O_2_ concentration during resuscitation.

## Figures and Tables

**Figure 1 children-09-01750-f001:**
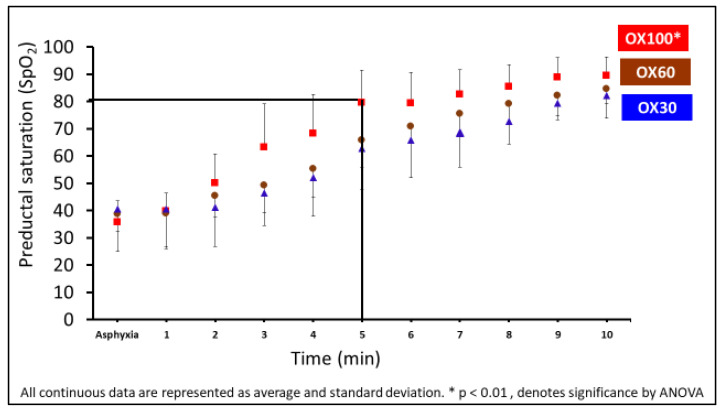
The graphs here shows the preductal saturations (SpO_2_) for OX30 (triangle), OX60 (circle), and OX100 (square) on the *y*-axis and the time from birth on the *x*-axis. The preductal SpO_2_ was significantly higher with OX100 than with OX30 in the first 10 min (* *p* < 0.01, denotes significantly by ANOVA). The black intersecting line shows the 5 min (*x*-axis) and 80% SpO_2_ mark (on *y*-axis). Data are represented as average and standard deviation.

**Figure 2 children-09-01750-f002:**
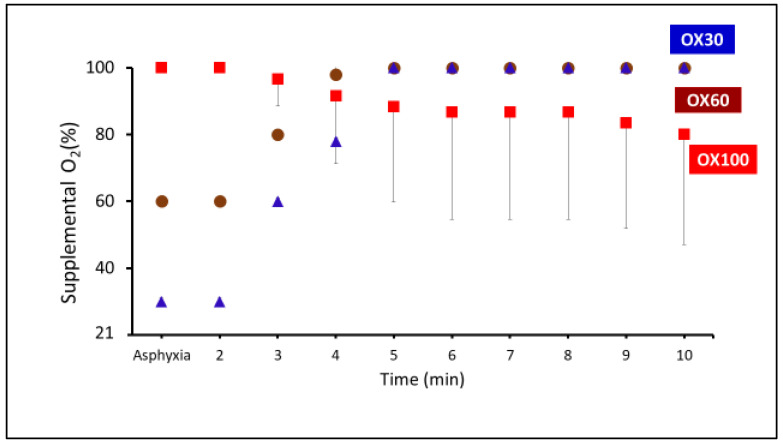
The graph represents the supplemental oxygen used during the resuscitation in OX30 (triangle), OX60 (circle), and OX100 (square) groups. The oxygen was titrated after 2 min. Data are represented as average and standard deviation.

**Figure 3 children-09-01750-f003:**
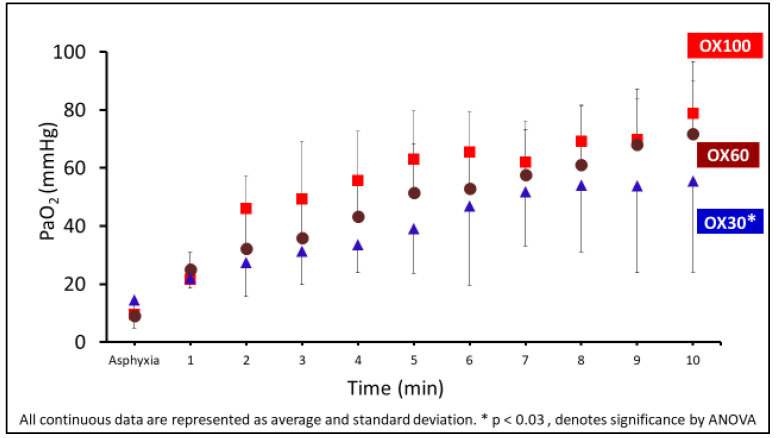
The graph shows the arterial oxygenation (PaO_2_) in mmHg on the *y*-axis and the events in min on the *x*-axis. The PaO_2_ was significantly lower in the OX30 (triangle) group (* *p* = 0.03 by ANOVA). Data are represented as average and standard deviation.

**Figure 4 children-09-01750-f004:**
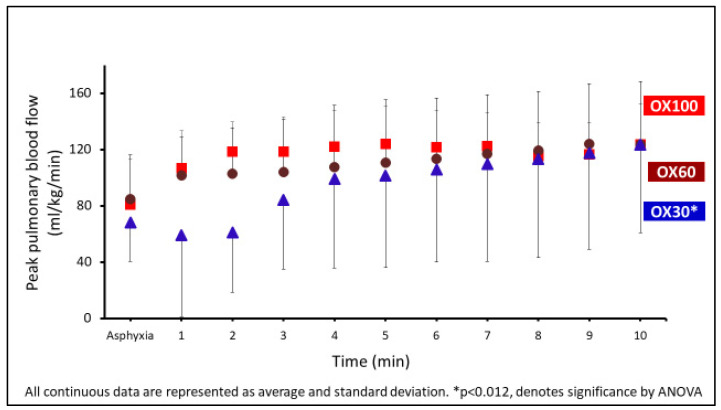
The graph shows the peak left pulmonary blood low (PBF) in mL/kg/min on the *y*-axis and the events in min on the *x*-axis. The PBF was significantly lower in the OX30 (triangle) group (* *p* = 0.012 by ANOVA). Data are represented as average and standard deviation.

**Figure 5 children-09-01750-f005:**
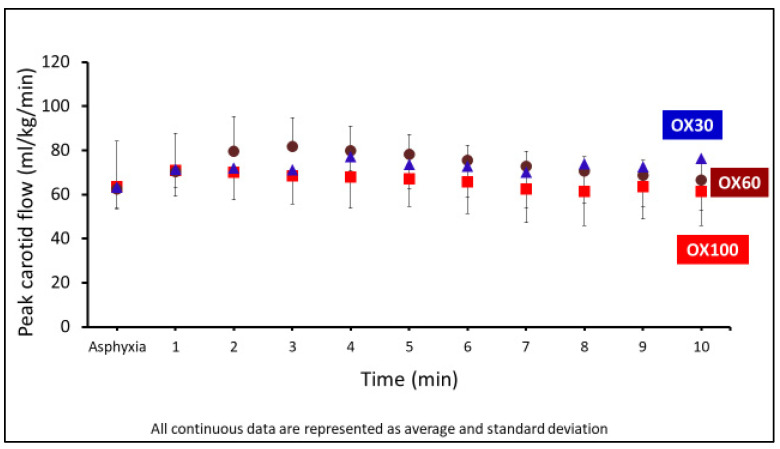
The graph shows the peak carotid blood low (CBF) (obtained from the left carotid artery) in mL/kg/min on the *y*-axis and the events in min on the *x*-axis. The CBF was not different between the groups. Data are represented as average and standard deviation.

**Figure 6 children-09-01750-f006:**
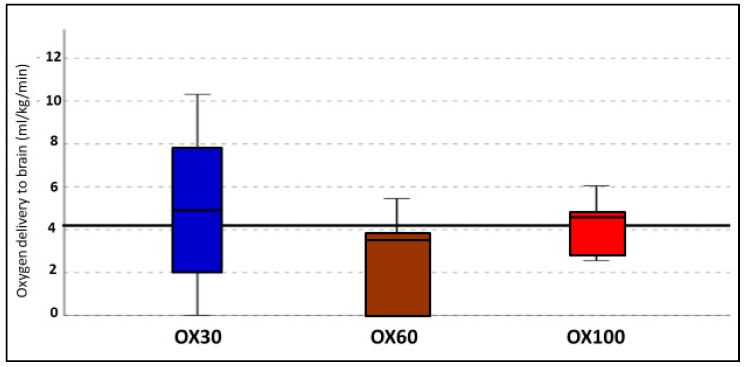
The oxygen delivery (*y*-axis) as mL/kg/min to the brain in different groups (OX30, OX60, and OX100 (*x*-axis)) are shown as box and whiskers plot. The horizontal black line plotted at 4 mL/kg/min is the median oxygen delivery in all the groups. There was wide variability in cerebral oxygen delivery with the use of OX30. There was no statistical difference between the groups.

**Figure 7 children-09-01750-f007:**
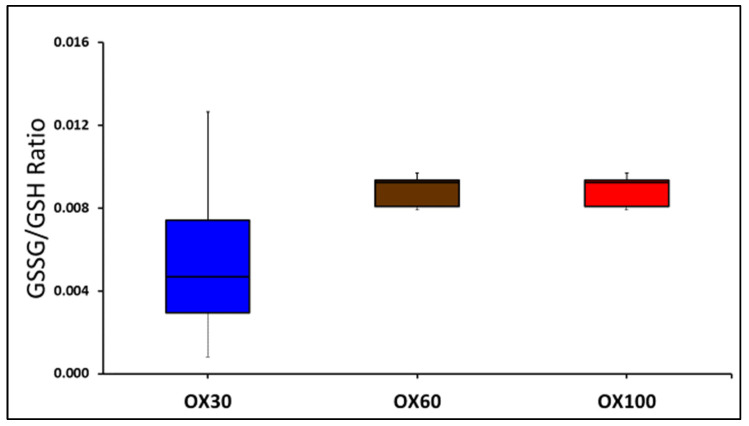
The oxidized to reduced glutathione ratio in the blood (*y*-axis) is shown between the three groups (OX30, OX60, and OX100) as box and whiskers plot. The markers of oxidative injury were not different between the groups at 10 min.

**Table 1 children-09-01750-t001:** Characteristics of preterm lambs.

Parameter	OX30	OX60	OX100
N	6	5	7
Gestational age (days)	127 ± 0.89	127 ± 0.84	126 ± 0.75
Female (N)	3	3	4
Multiplicity	Twins—2	Twins—3	Twins—3
pH at asphyxia	6.92 ± 0.17	6.91 ± 0.04	6.9 ± 0.18
HR at asphyxia (bpm)	91 ± 3	90 ± 3	88 ± 4

Data represented as mean and standard deviation.

**Table 2 children-09-01750-t002:** Primary outcome: combined HR of ≥100 bpm and SpO_2_ ≥ 80% by 5 min.

Parameter	OX30	OX60	OX100	*p*-Value
Percentage achieving primary outcome (%)	0	40%	67%	0.385
Time taken to achieve primary outcome (s)	600 ± 180	480 ± 180	360 ± 120 *	0.04

Data represented as mean and standard deviation. * *p* < 0.04, denotes significantly by ANOVA.

## Data Availability

The data presented here are only for the first ten min. Once the rest of the analysis have been completed, data will be available based on request.

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
