# Peer review of "Initial Use of 100% but Not 60% or 30% Oxygen Achieved a Target Heart Rate of 100 bpm and Preductal Saturations of 80% Faster in a Bradycardic Preterm Model"

_children, 2022, doi:10.3390/children9111750_

Round 1

Reviewer 1 Report

This manuscript is an ovine study wherein the authors provide neonatal resuscitation of an asphyxiated preterm lamb using three different starting oxygen concentrations (30, 60, and 100%) assessing the primary outcome of achieving goal heart rate and saturations by 10 minutes of life.  The experimental methods of the study are sound.  The presentation and interpretation of the data, however, require improvement before it is suitable for publication.  For that reason, significant revisions would increase enthusiasm for its publications.

MAJOR CRITIQUES
- All graphs should be plotted as individual data points with error bars (Figures 1, 2, 8, and 9)
- Figures 1 and 2 are unnecessary and can be presented in table format
- The more informative figures are the time-ordered changes in hemodynamic parameters
- Figure 4 - was the oxygen weaned if the lamb was in target range.  This is implied in the figure (and methods) but should be explicitly stated
- Median and ranges may be more informative than average and standard deviation with such a small N -- for example 89 +/-22% oxygen is conceptually confusing since the range can't exceed 100%
- If the data are continuous waveforms, why have the authors chosen to present them in only 1 minute intervals?
- Several statements are at risk of overstating the data -- suggesting there are differences when the groups are not statistically significant. For example:

1. "However, the corresponding PaCO2 in the groups, OX30 110±37, OX60 104±28, OX30 102±16) mmHg were higher but not different between the groups." -- They can't be "higher but not different"
2. "This discrepancy between ETCO2 & PaCO2 levels in this model was expected." -- That is not intuitive to this reviewer and deserves explanation
3. "In this ovine model, there were subtle differences noted in the three concentrations of O2 used during resuscitation on oxidized to reduced glutathione ratio in the first ten min." -- There can't be subtle differences if there is no statistical trend or change

- GSH/GSSG is a relatively crude marker of oxidative stress. Is there anything additional that can support that there is not undue oxidative stress in the OX60 and OX100 groups?

MINOR CRITIQUES
- Formatting requires additional attention (in appropriate italics on page 2 methods)
- Grammatical errors are present throughout
- Time is presented in minutes, but should be presented in minutes:seconds if they are known.
- Error bars on graphs seem to be unidirectional and at times overlapping making the interpretation of the graphs difficult

Author Response

Reviewer 1

This manuscript is an ovine study wherein the authors provide neonatal resuscitation of an asphyxiated preterm lamb using three different starting oxygen concentrations (30, 60, and 100%) assessing the primary outcome of achieving goal heart rate and saturations by 10 minutes of life.  The experimental methods of the study are sound.  The presentation and interpretation of the data, however, require improvement before it is suitable for publication.  For that reason, significant revisions would increase enthusiasm for its publications.

We thank the reviewer for the comments and have made changes as suggested below.

MAJOR CRITIQUES
- All graphs should be plotted as individual data points with error bars (Figures 1, 2, 8, and 9)

We have plotted the graphs as suggested by the reviewer as individual data points with error bars.

- Figures 1 and 2 are unnecessary and can be presented in table format.
- The more informative figures are the time-ordered changes in hemodynamic parameters

Thank you for the suggestion. We have changed figure 1 & 2 to table format.

- Figure 4 - was the oxygen weaned if the lamb was in target range?  This is implied in the figure (and methods) but should be explicitly stated.

Yes, if the target saturation was not achieved, as mentioned in the methodology, the supplemental oxygen was titrated (increased/weaned). This has been explicitly stated under the result section as per the reviewer’s recommendation.

- Median and ranges may be more informative than average and standard deviation with such a small N -- for example 89 +/-22% oxygen is conceptually confusing since the range can't exceed 100%.

In addition to the figure, we have also given the data as median and ranges in the text.

- If the data are continuous waveforms, why have the authors chosen to present them in only 1 minute intervals?

Since the study is based on titrating supplemental oxygen every min and to learn the subsequent effects, we chose to present the data in 1 min intervals.

- Several statements are at risk of overstating the data -- suggesting there are differences when the groups are not statistically significant. For example:

  1. "However, the corresponding PaCO2 in the groups, OX30 110±37, OX60 104±28, OX30 102±16) mmHg were higher but not different between the groups." -- They can't be "higher but not different"

We have corrected this sentence and wrote it’s not different.

  1. "This discrepancy between ETCO2 & PaCO2 levels in this model was expected." -- That is not intuitive to this reviewer and deserves explanation

In a surfactant deficient asphyxiated model, without antenatal betamethasone, there is much of carbon dioxide retention. With ventilation perfusion mismatch in the absence of surfactant, there was considerable difference between ETCO2 and PaCO2 existed in our model. This sentence has been clarified in the paper.

  1. "In this ovine model, there were subtle differences noted in the three concentrations of O2 used during resuscitation on oxidized to reduced glutathione ratio in the first ten min." -- There can't be subtle differences if there is no statistical trend or change

We have removed this sentence and have mentioned it as ‘no differences noted’.

- GSH/GSSG is a relatively crude marker of oxidative stress. Is there anything additional that can support that there is not undue oxidative stress in the OX60 and OX100 groups?

Since GSH/GSSG ratio is present in mammalian cells, we chose this as a close marker of oxidative stress in our study. Future studies will have more markers and staining methods to assess the oxidative stress using various oxygen concentration.

MINOR CRITIQUES
- Formatting requires additional attention (in appropriate italics on page 2 methods)

  Italics were used to highlight the oxygen titration. Now it has been formatted to the regular font.

- Grammatical errors are present throughout

   We have carefully read the paper and corrected the errors.

- Time is presented in minutes, but should be presented in minutes:seconds if they are known.

  We have changed the time to seconds.

- Error bars on graphs seem to be unidirectional and at times overlapping, making the interpretation of the graphs difficult

A bidirectional error bar, makes it even more difficult to interpret data and hence left it unidirectional.

Reviewer 2 Report

This study is very well done, sound and very relevant. 

The median oxygen delivery to the brain in all three groups(OX30,OX60 and OX100) were not significantly different though based on data presented in figure 8, there is wide variation in OX30 group. Would the authors give a possible explanation for this effect?

Thank you for the opportunity to review this paper. 

Author Response

Reviewer 2

This study is very well done, sound and very relevant. 

The median oxygen delivery to the brain in all three groups (OX30,OX60 and OX100) were not significantly different though based on data presented in figure 8, there is wide variation in OX30 group. Would the authors give a possible explanation for this effect?

Thank you for the opportunity to review this paper. 

Thank you for the encouraging words. In the figure 8, which is now figure 6, the distribution of OX30 was near normal, meaning the median and the interquartile ranges were equally distributed compared to OX60 and OX100 where the median was closer to the 75th interquartile range. As the reviewer pointed out, the variation could be secondary to the fact that lower supplemental oxygen could have led to a variability in cerebral oxygen delivery in an asphyxiated surfactant deficient model. This has been added to the discussion.

Reviewer 3 Report

I wish to thank the authors for this study which is very well written, findings clearly presented and conclusion sound and based on the findings in their study.  I do think this study highlights the need for additional testing in preterm neonates to determine the optimal FiO2 to begin resuscitation with.  It would be very helpful to more fully delineate the oxidative stress caused by 100% in premature neonates as compared to term neonates.

Author Response

Reviewer 3

I wish to thank the authors for this study which is very well written, findings clearly presented and conclusion sound and based on the findings in their study.  I do think this study highlights the need for additional testing in preterm neonates to determine the optimal FiO2 to begin resuscitation with.  It would be very helpful to more fully delineate the oxidative stress caused by 100% in premature neonates as compared to term neonates.

Thank you for your kind words. We are working on more oxidative stress markers to understand the differences between premature and term neonates with exposure to 100% oxygen. We have added this to our limitations.

Round 2

Reviewer 1 Report

Thank you for addressing many of the concerns and improving the clarity of the figures in the paper. 

Major:

Figures 6 (O2 delivery to the brain) and 7 (GSSG/GSH ratios) show box and whisker plots, not individual data points as requested in the first round of reviews.  Recommend updating those figures before publication.

Minor:

Typo at beginning of discussion "stray ] closed bracket"

Table 2 should have a p value for both rows (the primary outcome and time to achieve primary outcome)

The abstract should explicitly state whether the primary outcome reached statistical significance -- as the conclusion does

Author Response

October 20, 2022

To,

The Editor,

Children MDPI.

Re: Children-1946575 - Initial use of 100% but not 60% or 30% oxygen achieved a target heart rate of 100 bpm and preductal saturations of 80% faster in a bradycardic preterm model

Dear Editor,

We thank you and the reviewer for the comments and suggestions. The changes made in response to these comments have substantially improved the quality of the manuscript. Kindly find our response to reviewer comments below:

Reviewer 1:

Thank you for addressing many of the concerns and improving the clarity of the figures in the paper. 

Major:

Figures 6 (O2 delivery to the brain) and 7 (GSSG/GSH ratios) show box and whisker plots, not individual data points as requested in the first round of reviews.  Recommend updating those figures before publication.

For these two data, we cannot do a minute to minute box and whisker plot as requested by the reviewer for the following reasons:

Our goal is to minimize blood draw during the experiments. The whole blood sample collected at 10 min was used to calculate the GSSG/GSH ratio and the hemoglobin levels using which we did calculate the oxygen delivery to the brain (arterial oxygen content). Hence, these values are furnished only at 10 min. We have previously mentioned in the methodology, that the sample for GSSG/GSH was collected at 10 min in the methodology and now we have also added that this sample was used to calculate oxygen delivery to the brain.

Minor:

Typo at beginning of discussion "stray ] closed bracket"

Thank you, we have removed it.

Table 2 should have a p value for both rows (the primary outcome and time to achieve primary outcome)

P value has been added to both the rows.

The abstract should explicitly state whether the primary outcome reached statistical significance -- as the conclusion does

This has been corrected.

We thank the reviewer and the editor again and hope the changes made is acceptable for publication.

Sincerely,

Praveen Chandrasekharan
